behaviour/cognition/evolution

primate vocal communication, predator recognition, call comprehension, meaning attribution, playback experiment, fast mapping

**Author for correspondence:**
Adwait Deshpande
e-mail: adwait.deshpande2390@gmail.com

# Preliminary evidence for one-trial social learning of vervet monkey alarm calling

Adwait Deshpande[1,2], Bas Van Boekholt[2,3,4] and Klaus Zuberbuhler[1,2,5]

[1]Department of Comparative Cognition, Institute of Biology, University of Neuchatel, Neuchatel, Switzerland
[2]Inkawu Vervet Project, Mawana Game Reserve, KwaZulu-Natal, South Africa
[3]Animal Ecology, Department of Biology, Utrecht University, Utrecht, The Netherlands
[4]Comparative BioCognition, Department of Cognitive Science, University of Osnabrück, Germany
[5]School of Psychology and Neuroscience, University of St Andrews, St Andrews, UK

AD, 0000-0003-2581-2737

How do non-human primates learn to use their alarm calls? Social learning is a promising candidate, but its role in the acquisition of meaning and call usage has not been studied systematically, neither during ontogeny nor in adulthood. To investigate the role of social learning in alarm call comprehension and use, we exposed groups of wild vervet monkeys to two unfamiliar animal models in the presence or absence of conspecific alarm calls. To assess the learning outcome of these experiences, we then presented the models for a second time to the same monkeys, but now without additional alarm call information. In subjects previously exposed in conjunction with alarm calls, we found heightened predator inspection compared to control subjects exposed without alarm calls, indicating one-trial social learning of 'meaning'. Moreover, some juveniles (but not adults) produced the same alarm calls they heard during the initial exposure whereas the authenticity of the models had an additional effect. Our experiment provides preliminary evidence that, in non-human primates, call meaning can be acquired by one-trail social learning but that subject age and core knowledge about predators additionally moderate the acquisition of novel call-referent associations.

# 1. Introduction

A vexing problem in animal communication research concerns the evolutionary discontinuities in vocal flexibility. On the one hand, songbirds, several cetaceans and some bats are on par with humans in terms of vocal learning abilities [1], whereas our

closest evolutionary relatives, the non-human primates, including the great apes, communicate with astoundingly small and acoustically inflexible vocal repertoires [2]. An important point in this discussion is that limitations in vocal production do not equate to limitations in comprehension and usage. For example, some simian species use various alarm call types and their combinations flexibly to refer to different contexts [3–7]. Moreover, chimpanzees (*Pan troglodytes*) produce alarm calls to actively inform ignorant group members of danger, suggesting that vocal production is tied to complex mental representations that, in the case of chimpanzees, may even involve control over call production [8,9]. Also relevant is that non-human primates sometimes use the same call type in multiple contexts, such as during predator encounters or between-group aggression, requiring receivers to make pragmatic judgements [10,11]. Although there is little doubt about the underlying cognitive richness of primate communication, it is still largely unclear how individuals actually acquire such competence.

In animal communication, questions concerning cognition have been addressed with research on alarm calls, not because alarm calls are thought to involve especially complex cognition but simply because the referents are external, which makes them easier to manipulate experimentally than the referents of social interactions. Regarding theoretical frameworks, one possibility is that alarm calling is mainly hard-wired, i.e. developing without much environmental input. This is highly unlikely because the range of relevant referents (i.e. predator types) and environmental conditions is habitat-specific and too large for any kind of hard-wired system to evolve [12]. A second way to acquire competence in alarm calling is through individual trial-and-error learning. However, committing errors during predator encounters can be costly, and it is unclear what kind of feedback would foster novel call-context associations. Finally, a third possibility is social learning from observing conspecifics [13]. Predation pressure often changes over time and space, with novel predators suddenly appearing or existing ones disappearing (e.g. [14]), suggesting that social learning is a very efficient mechanism to acquire and maintain the necessary competence [15–17]. In line with this, social learning has already been established as an important mechanism for predator recognition and avoidance in other taxa (reviewed in [15]). For example, in a classic study, European blackbirds began to mob an innocuous bird model or even plastic bottles when observing other conspecifics doing the same [18].

Although the role of social learning has been explored in non-human primates, the current literature is mainly concerned with how animals acquire arbitrary food preferences or new manipulative skills [19,20]. Yet, learning from conspecifics is likely to be important in all biological domains, including communication. However, current taxonomies of social learning have been developed mainly to distinguish how animals acquire manipulation skills (e.g. local enhancement, emulation, imitation) but are less useful in other domains of behaviour, i.e. acquisition of communication competence or knowledge about world facts more generally. Regarding vocal learning itself [21], several processes have been distinguished, although the roles of individual and social learning have not been specifically addressed. First, in comprehension learning, individuals learn to associate a novel context with an existing signal, e.g. a predator class to an acoustically distinct alarm call. Second, in usage learning [21], individuals learn to produce an existing signal in a new situation, for example, as a result of watching other individuals doing it.

A few early studies have investigated socially acquired predator recognition in primates, mainly under captive conditions [22,23], with little known about the wild (but see [24]). An important but hitherto unresolved problem is how competence in comprehension and usage learning actually emerges [25]. Seyfarth & Cheney have proposed an influential model for the development of call use in vervet monkeys [26], which suggests that infants are equipped with the ability to produce adult-like alarm calls from early on, but that they need to learn the correct use of these calls. In particular, aerial-like alarm calls are given from the beginning, albeit first to a broad range of flying or falling objects, including non-predatory birds, suggesting some core knowledge about basic predator types or event categories and associated alarm calls (see [27]). In the next step, within-category use gets refined, such that juveniles produce aerial alarm calls only for specific predator classes (e.g. martial eagles *Polemaetus bellicosus*) whereas non-dangerous flying animals no longer elicit calls. Anecdotally, this process is driven by social learning, based on observations that juveniles who looked at adults before responding tended to respond correctly [26,28].

We attempted to address the learning process by experimentally testing whether conspecific alarm calls facilitate social learning in wild vervet monkeys and whether this can lead to new call-context associations and call use. South African vervet monkeys (*Chlorocebus pygerythrus pygerythrus*) are an ideal model system for such investigations. The classic studies on the Kenyan sub-species (*Chlorocebus pygerythrus hilgerti*) showed that adults produced at least three acoustically distinct alarm calls for corresponding predator categories (aerial threats, terrestrial predators and snakes). The different alarm

calls evoked predator-specific responses, such as running into the bush for aerial threats, climbing up trees for terrestrial predators, and standing bipedally to scan for snakes [29,30]. A recent study on the South African sub-species revealed, however, that adults also used alarm calls in non-predatory contexts, particularly during intra- and inter-group aggression [31].

We capitalized on this well-studied system and exposed groups of free-ranging South African vervet monkeys to two novel animal models (commercially available soft toys for toddlers). In a cross-experimental design, we provided additional information about the model during the first encounter. We did this by broadcasting alarm calls or non-alarm calls by a known adult group member or by not playing any call at all. We predicted that if comprehension learning took place, individuals from the informed group should show increased predator inspection during any subsequent encounters with the model, compared to individuals from a control group. We also predicted that socially acquired comprehension would lead to competence in call usage, such that individuals from the informed group should be more likely to produce alarm calls during subsequent encounters with the model. We took into account the effects of the predator model type and subject age, since both variables could impact acquisition.

# 2. Methods

## 2.1. Study site and groups

The study was conducted at the Inkawu Vervet Project (IVP) on the Mawana Game reserve, Kwazulu Natal, South Africa (28°00.327 S; 031°12.348 E) from November 2018 to July 2019. At the IVP, six groups of wild vervet monkeys are well habituated to human presence and allow close observations by multiple observers. All the observers undergo standard training and pass a test for individual identification and inter-observer reliability before participating in data collection. Individuals were fed regularly on corn kernels for various other behavioural experiments. Food provisioning events were announced using specific calls by observers, which are understood by the monkeys and attract them to the location of the food [20,32,33]. The experiments presented here were conducted on three different groups: (i) the BD group ($n = 29$ adults, $n = 18$ juveniles); (ii) the AK group ($n = 5$ adults, $n = 14$ juveniles); and (iii) the NH group ($n = 17$ adults, $n = 20$ juveniles). As per IVP protocol, females were considered juveniles until they gave birth to their first infant; males were considered juveniles until they migrated from their natal group, usually on or before 4 years of age.

## 2.2. Experimental set-up

We conducted four blocks of experiments on the two groups of monkeys (BD, AK). Each block consisted of two parts: group exposures followed by individual assessment trials. Group exposures consisted of four different treatments; two animal models (caterpillar versus horse, see below) by two information states (alarm call playback versus no playback). In the initial design, we also had a third information state as a further control condition, grunt call playback), but owing to a technical problem, data collection was interrupted midway, so we can only report a fraction of the results (see below). For the remaining conditions, we cross-balanced the treatments of alarm playback (Alarm) and no playback (Silent control) with the two animal models (table 1).

To ensure the playback experience was perceived as plausible for subjects, we used recordings of alarm calls from an adult male of the same group, produced naturally for approaching jackals or feral dogs, which are common terrestrial mammalian predators in the study area. In the 'Grunt control' condition (see experimental design), we used recordings from a subordinate adult male vocalising to the dominant males of the same group [34]. As a result, each group was exposed to a different playback stimulus during the exposure trials (Alarm treatment). Acoustically, the alarm call playbacks were very similar to the 'leopard' alarm calls of Amboseli vervet monkeys and henceforth labelled as such for convenience [30]. Playback stimuli comprised 10–14 leopard alarm calls at natural intervals clustered into 3–4 bouts. Stimuli were presented with a Nagra Kudelski DSM-Monitor speaker played through a Huawei ALE-L21 phone. The speaker was hidden under a blanket as well as behind a bush, approximately 10–15 m away from the cylindrical hide of the models (see below). We pre-adjusted speaker amplitude to match the perceived amplitude of a natural call at the relevant distance. Original stimuli were edited such that the amplitude of other background sounds was reduced. All the call tracks were padded at both ends by 2 s of silence using the software AUDACITY 2.3.3 [35].

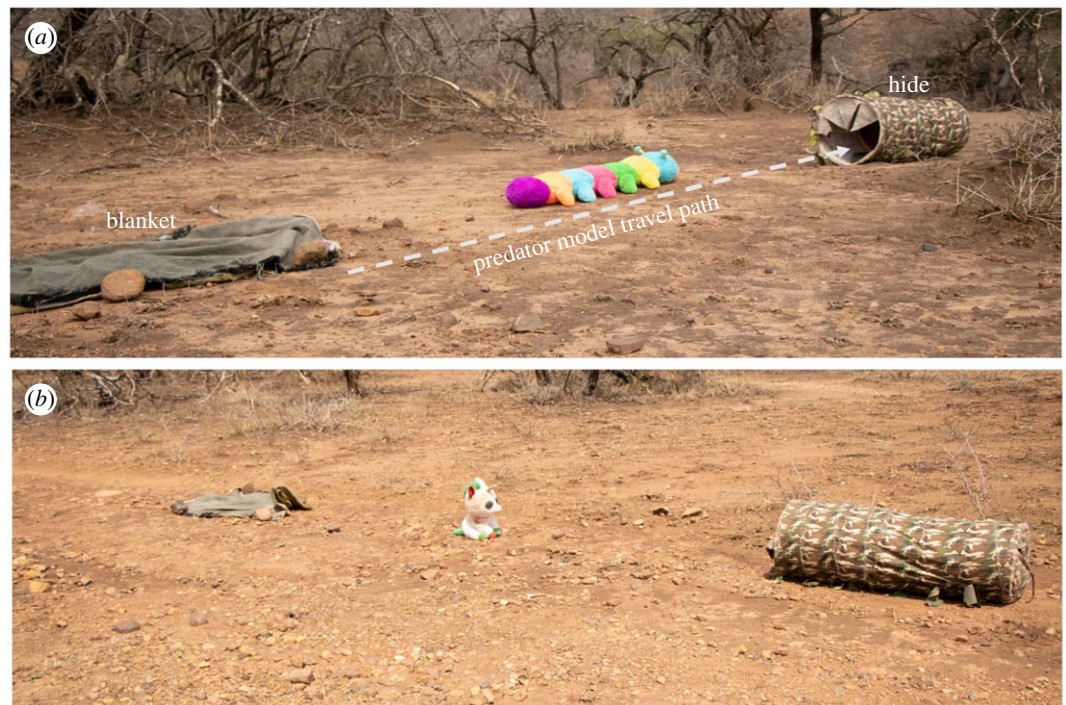

**Figure 1.** Experimental set-ups showing two different predator models: (*a*) caterpillar, (*b*) horse.

**Table 1.** Sample size for individual assessment trials across treatments for the study groups AK, BD and NH.

| treatment | predator | sample size ($n$) for individual assessments for each study group |
|---|---|---|
| Silent control (no playback) | caterpillar | AK $n = 6$ |
| Alarm (leopard alarm playback) | caterpillar | BD $n = 8$ |
| Alarm (leopard alarm playback) | horse | AK $n = 6$ |
| Silent control (no playback) | horse | BD $n = 6$ |
| Grunt control (grunt playback) | caterpillar | NH $n = 5$ |

We were interested in how novel call-context associations are formed and how this could lead to competence in comprehension and production of alarm calls. For this, we used unfamiliar animal-like models as stimuli, completely alien to the monkeys. We chose two commercially available soft toys for toddlers: (i) an artificially coloured, oversized caricature of an insect larva (caterpillar); and (ii) a miniaturized caricature of an ungulate ('horse'; figure 1). Models were comparable in size but differed in terms of animacy and resemblance to real animals. When pulled, the 'caterpillar' model made an animate movement with mobile legs, suggesting acceptable plausibility as an animate object, whereas the 'horse' model toppled sideways and could only be dragged inanimately, suggesting low authenticity despite its conspicuousness.

The experimental arena consisted of a blanket and a camouflaged cylindrical hide, placed approximately 10 m apart on the ground. The predator model was hidden under the blanket, attached with a fishing line and controlled by the experimenter, who was positioned at least 2 m behind the hide. A trial began with the experimenter pulling the predator model from under the blanket and dragging it in a straight line across the arena to disappear inside the cylindrical hide. A tripod with a

(a)

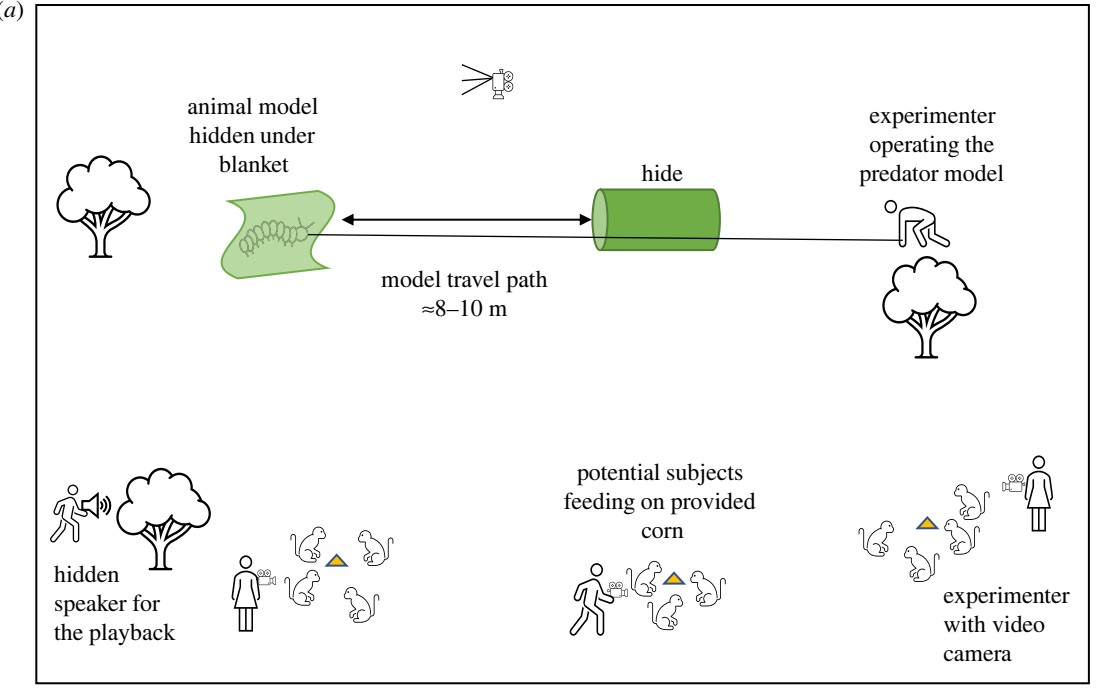

(b)

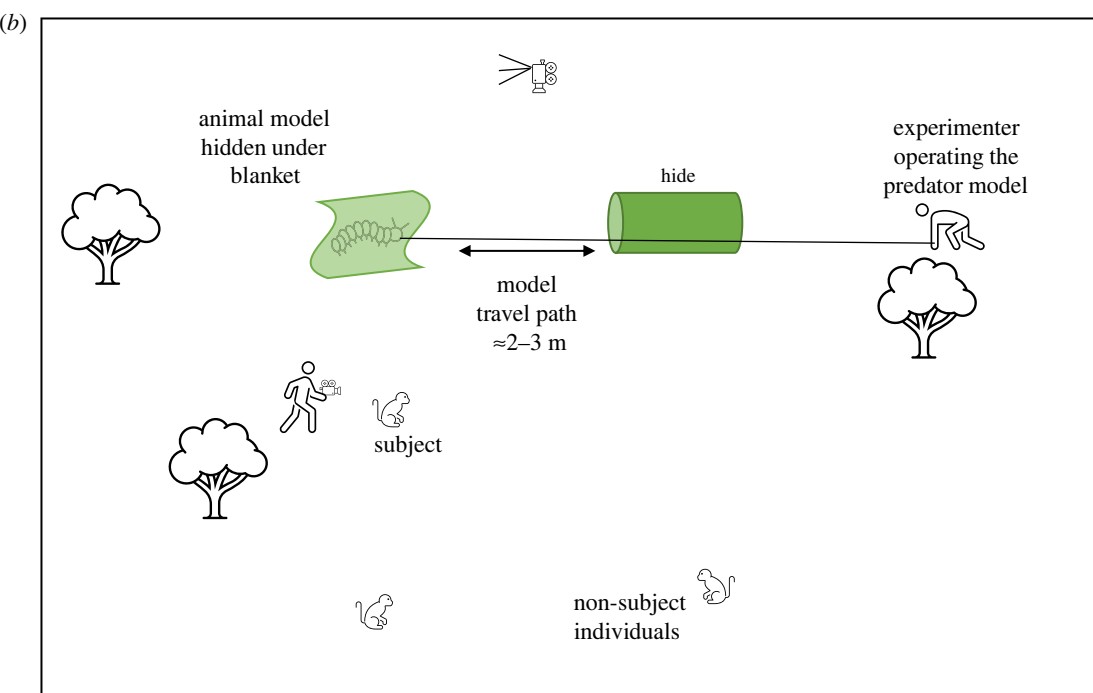

**Figure 2.** Schematic illustrations (not to scale) of experimental set-ups for different trials: (a) group exposure, (b) individual assessment.

video camera was placed to record all events in the arena (figure 2a). We accustomed both groups to the experimental arena (without models) by repeatedly (5–8 times) placing the set-up in the group's travelling path until all individuals ignored the set-up.

## 2.3. Experimental procedure

An experimental block started with a group exposure on the first day, followed by individual assessment trials on subsequent days (electronic supplementary material, figure S1). Group exposures were conducted after the monkeys left their sleeping sites in the morning. In advance, we set up the experimental arena on the group's most likely travel path and attracted individuals with food calls to

small mounds of corn kernels placed 5–10 m away from the set-up. For the experimental (Alarm) treatment, once most of the monkeys were present at the mounds, we played the alarm call stimulus from the hidden speaker and simultaneously moved one of the models across the arena as described above (figure 2a). The exposure lasted approximately 20–30 s. For the Silent control treatment, we did precisely the same, except there was no playback. All group exposure trials were recorded on video using a JVC quadproof EverioR camcorder. The experimenter was assisted by 2–4 observers who identified individuals present during the first seconds of exposure and recorded which individuals saw the model by coding head orientation towards the model. These individuals were considered as 'informed' and thus became subjects for the subsequent assessment trials. Twenty-six individuals opportunistically qualified for subject status across the treatments (adult females: $n = 4$; adult males: $n = 4$; juvenile females: $n = 8$; juvenile males: $n = 10$). Once a model was no longer visible, individuals often returned to foraging on the remaining food, usually after approximately 10–15 min.

In each block, we conducted one assessment trial per subject per day, starting on the day after a group exposure. This design was adapted to minimize the confounding effects of natural predator encounters and other natural alarm call events. The frequency of natural terrestrial predator encounters at the study site varied significantly from once per day to once per week (A. Deshpande 2018, unpublished data). During individual assessment trials, we set up the arena and presented the model in the same way, but this time only to one subject, with no playback and reduced exposure time (mean 9.6 s; s.e. ± 1.1 s) by reducing the distance between the blanket and the hide from 10 m to about 2.5 m (figure 2b). We exposed the model only when the subject looked in the general direction of the blanket, to ensure the model was easily noticed by the subject (see the electronic supplementary material, videos). For one trial, a model was not noticed, and we hid the model in the cylindrical hide immediately. A re-trial on the same subject was conducted later on the same day. Individual assessment trials on the subjects were conducted opportunistically, ensuring that no other subject was in the vicinity at the time of the trial. For each trial, we noted the distance between the subject and the experimental arena. Subjects were recorded for 10 min, starting with the model exposure. We chose a longer time window to monitor the behaviours because in a pilot exposure to another group, the monkeys inspected and alarm called at the hide for more than 5 min. During one of the trials, the observer lost the subject within two minutes of exposure; hence, this trial was excluded from the final analyses. We also audio-recorded all alarm calls and other vocalizations elicited by the subject using a solid-state audio recorder (Marantz PMD 661) and a directional microphone (Sennheiser MKH 416 P48) with a sampling frequency of 44.1 kHz.

## 2.4. Experimental design

The original study design included three treatments ('Alarm', 'Silent control', 'Grunt control'), administered separately to the three groups (NH, AK, BD) with two predator models ('horse', 'caterpillar'; figure 3). However, the speaker malfunctioned during the key NH group exposure, to the effect that the NH group was exposed to the 'horse' model without the necessary 'Alarm' treatment. As this experience was critical for the NH individuals, we were forced to adjust the experimental design by assigning the 'horse' model/'Alarm' treatment to the AK group (instead of the initially planned 'Grunt control' treatment). As this created an imbalance, we removed the 'Grunt control', treatment and compared the two key treatments ('Alarm' versus 'Silent control'). It is obvious that this represents an unfortunate limitation of our study, caused by the vagaries of fieldwork. Nevertheless, we decided to present the results of the remaining 'Grunt control' treatments as a subsidiary analysis for comparison (see statistical analysis).

## 2.5. Behaviour coding

All individual assessment videos were coded using the video coding software BORIS v. 6.3.9 [36]. We coded all the behaviours elicited during the first 10 min from exposure to the model. We were mainly interested in the following two behaviours: (i) predator inspection (s)—total time spent looking directly at the predator during exposure and/or time spent looking directly at the cylindrical hide after the exposure; and (ii) number of alarm calls (n).

To assess the inter-observer reliability of coding, a second rater, experienced in primate behaviour but blind to the aims and objectives of the study, recoded 20% of the videos, which were randomly selected. For predator inspection (a continuous variable), we found a strong Spearman correlation ($\rho = 0.83$, $p = 0.015$), suggesting that data were coded in sufficiently comparable ways.

planned experimental design

| model / group | caterpillar | horse |
|---|---|---|
| BD | Alarm | Silent control |
| AK | Silent control | Grunt control |
| NH | Grunt control | Alarm |

adjusted experimental design

| model / group | caterpillar | horse |
|---|---|---|
| BD | Alarm | Suilent control |
| AK | Silent control | ~~Grunt control~~ alarm |
| NH | Grunt control | alarm (failed) |

**Figure 3.** Figure shows original and adjusted experimental designs with assigned treatments for the three groups of monkeys and two models. Owing to speaker malfunction during 'Alarm' treatment for the NH group for the 'horse' model, we had to adjust the experimental design. We conducted the same treatment for the AK group instead of the originally assigned 'Grunt control' treatment, creating an imbalance in the design.

## 2.6. Acoustic analysis and classification

We extracted a subject's alarm calls during the individual assessments from the audio recordings using AUDACITY 2.3.3 [35] for acoustic analyses after cross-checking them on the videos. Overlapping calls were removed from further analyses. We also included previously recorded alarm calls by A.D. These calls were recorded during natural predator encounters of jackals (Canis mesomelas), pythons (Python sebae natalensis) and different species of raptors and corresponded to leopard, snake and eagle alarms, respectively, as described in earlier studies and henceforth labelled as such (electronic supplementary material, figure S2) [29–31]. A call bout was defined as a coherent acoustic utterance, separated by at least 500 ms from another. Alarm call bouts could contain several calls separated by less than 500 ms from each other, marked in RAVEN PRO 1.5 [37] using Hamming window at 1024 DFT and 93.8% overlap. Acoustic parameters were extracted at the call level and submitted for subsequent statistical modelling. We extracted a total of 117 calls from 48 call bouts and initially extracted 23 acoustic parameters from each call.

To avoid potential coding biases, we used linear discriminant analysis (LDA) to classify the calls elicited by the subjects during the individual assessments. First, we scaled the data and reduced the dimensionality by removing highly correlated (greater than 0.95) acoustic parameters and used the remaining 16 parameters for the LDA (electronic supplementary material, table S1). Naturally recorded alarm calls ($n = 40$) were used as training data for the LDA model with equal prior probabilities. We then predicted the call types for each call recorded in the individual assessment trials based on the LDA model. LDA was performed using the lda function in the package MASS [38] function in R v. 3.6.3 [39].

## 2.7. Statistical analyses

We fitted generalized linear models with the lme4 package [40] to test the effects of the 'Alarm' and 'Silent control' treatments on the two behaviours elicited during individual assessments, predator inspection (s) and a total number of alarm calls ($n$) [41–43]. For predator inspection, we log-transformed the data to use a linear model; for alarm calls, we accounted for zero inflation and used negative binomial models. We controlled for type of predator model, group identity, age of subject (juvenile or adult) as categories; and the order of trial, the distance of the subject from the predator at the start of exposure and exposure time as Z transformed variables by adding them as fixed effects in all the models. To test for the significance of each fixed effect, we used the analysis of variance or deviance depending on the type of full model. We visually checked for heteroscedasticity and then verified that variance-inflation factors were less than four for all the fixed effects in the full

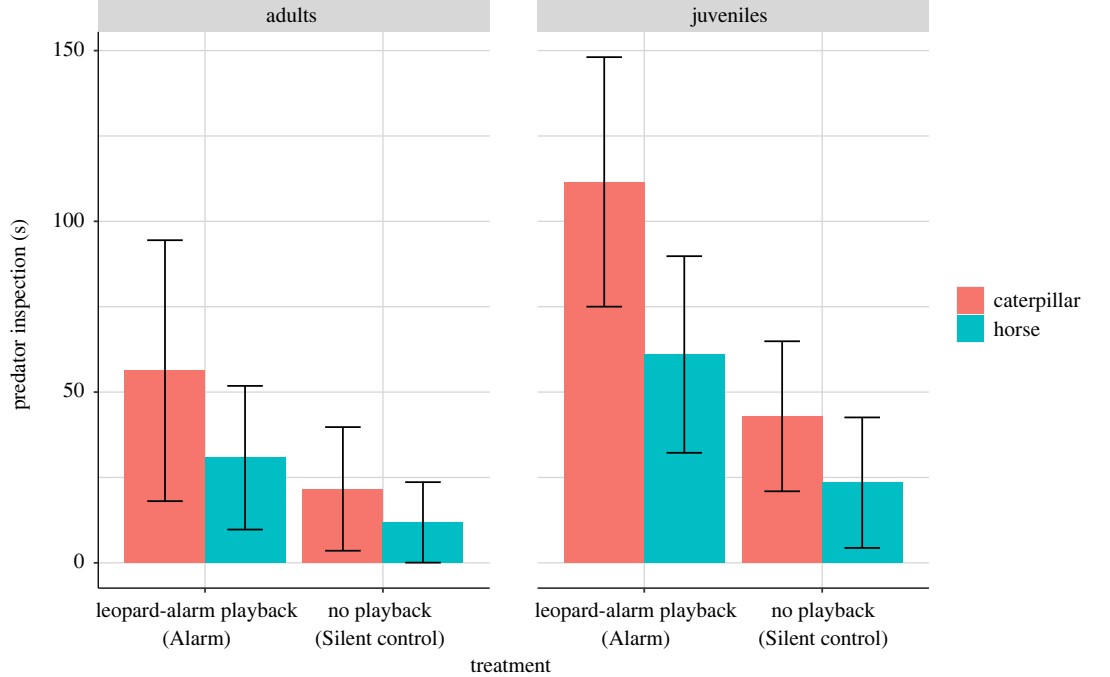

**Figure 4.** Model estimates (±s.e.) of the effects of treatments on the predator inspection of the subjects grouped by age categories and predator model type.

models [44]. Effect sizes were determined using the MuMIn package [45]. To assess the effects of the 'Alarm' treatment on alarm calls produced by juveniles to the 'caterpillar' model, we did a post hoc analysis by fitting negative binomial models on the number of leopard and snake calls on a corresponding subset of data. The fixed effects were the same as earlier models except for the predator model, and group identity were removed as fixed effects.

We ran a separate non-parametric analysis to compare the difference between 'Alarm' and 'Grunt control' treatments with Wilcoxon tests to address the possibility that changes in predator inspection could be caused by any vocalization uttered during the model exposure. Finally, the 'Grunt control' treatment did not lead to any alarm calls, so no further comparisons were made. All analyses were conducted in R v. 3.6.3 [39].

# 3. Results

## 3.1. Predator inspection

In the assessment trials, monkeys inspected both models for significantly longer following 'Alarm' compared to 'Silent control' treatments, irrespective of age ($n = 26$, $F_{1,18} = 8.36$; $p = 0.009$, $R^2 = 0.39$; figure 4). There was a trend of the 'caterpillar' model triggering longer inspection than the 'horse' model, but this was not statistically significant ($n = 26$, $F_{1,18} = 3.91$; $p = 0.063$). In a subsidiary analysis, we also compared predator inspection following 'Alarm' and 'Grunt control' treatments which again revealed that predator inspection was significantly longer in the 'Alarm' than 'Grunt control' treatments ($n = 19$, $W = 12$, $p = 0.033$; Wilcoxon rank-sum test, electronic supplementary material, figure S3).

## 3.2. Alarm calls

During individual assessments, subjects produced $n = 77$ alarm calls. $n = 37$ were classified as snake alarms, $n = 3$ as eagle alarms and $n = 37$ as leopard alarms (LDA; table 2). No alarm calls were produced during the individual assessment trials of the 'Grunt control' treatment. Regarding the total number of alarm calls we found no effect of treatment but differences in model type and subject age: the 'caterpillar' elicited more alarms than the 'horse' model and juveniles produced more alarms than

**Table 2.** Descriptive statistics of the behaviours for the data collected during individual assessment trials across treatments and age categories.

| treatment and sample size (n) of individual assessments per treatment | animal model | predator inspection (s) mean ± s.e. sample size (n) standard deviation (s.d.) | | leopard alarms (counts) mean ± s.e. standard deviation (s.d.) | | snake alarms (counts) mean ± s.e. | | eagle alarms (counts) mean ± s.e. | | total alarms (counts) mean ± s.e. standard deviation (s.d.) | |
|---|---|---|---|---|---|---|---|---|---|---|---|
| | | juvenile | adults | juvenile | adults | juvenile | adults | juvenile | adults | juvenile | adults |
| Alarm (leopard playback) n = 14 | caterpillar | 114.21 ± 26.7, n = 7, s.d. = 70.9 range = 19.2–228.1 | 20.27 n = 1 | 2.71 ± 1.0 s.d. = 2.8 range = 0–7 | 0 | 1.85 ± 0.6 | 0 | 0.14 ± 0.1 | 0 | 4.71 ± 1.4 s.d. = 3.9 range = 0–10 | 0 |
| | horse | 42.17 ± 11.3 n = 4, s.d. = 22.6 range = 29.2–76 | 66.9 ± 37.7 n = 2, s.d. = 53.3 range = 29.2–104.7 | 1.50 ± 0.5 s.d. = 1 range = 0–2 | 0 | 1.75 ± 1.1 | 0 | 0.25 ± 0.2 | 0 | 3.50 ± 1.7 s.d. = 3.4 range = 0–8 | 0 |
| Silent control (no playback) n = 12 | caterpillar | 40.18 ± 11.5 n = 5, s.d. = 25.8 range = 11.7–69.7 | 52.2 n = 1 | 1.8 ± 1.0 s.d. = 2.3 range = 0–6 | 1 | 2.20 ± 0.8 | 4 | 0 | 1 | 4.00 ± 1.4 s.d. = 3.1 range = 0–8 | 6 |
| | horse | 28.53 ± 9.2 n = 2, s.d. = 12.9 range = 19.4–37.6 | 18.13 ± 6.5 n = 4, s.d. = 13, range = 6.6–34.5 | 1.0 ± 1.0 s.d. = 1.4 range = 0–2 | 0 | 1.00 ± 1.0 | 0 | 0 | 0 | 2.00 ± 2.0 sd = 2.8 range = 0–4 | 0 |

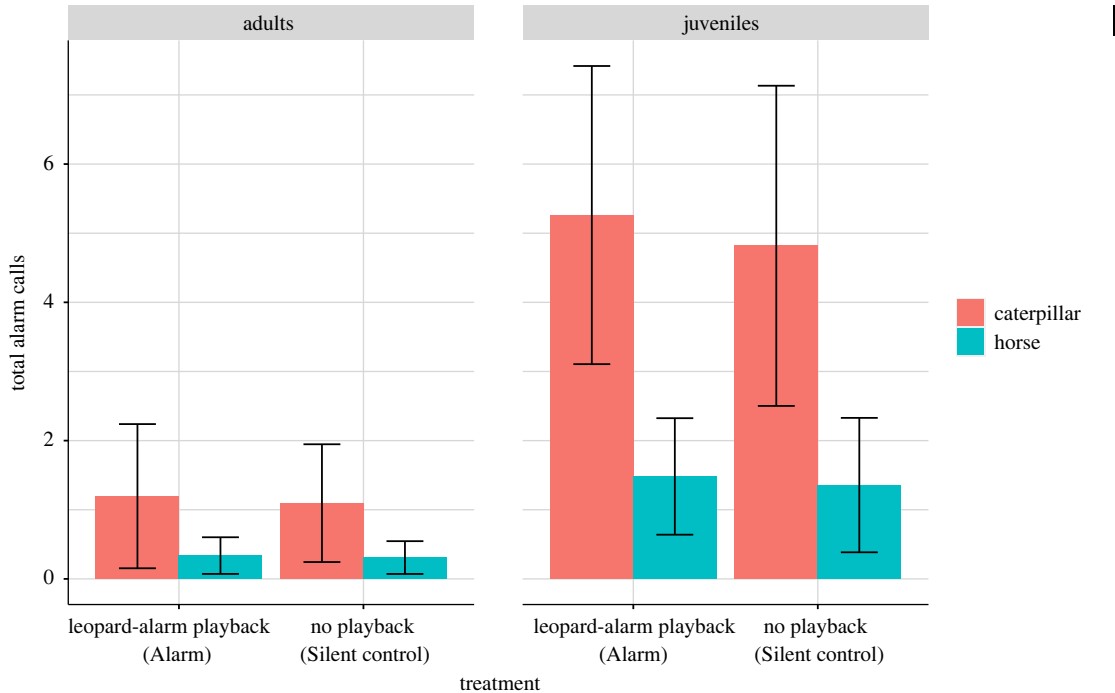

**Figure 5.** Model estimates (±s.e.) of the effects of treatments on total alarm calls of the subjects grouped by age categories and model type.

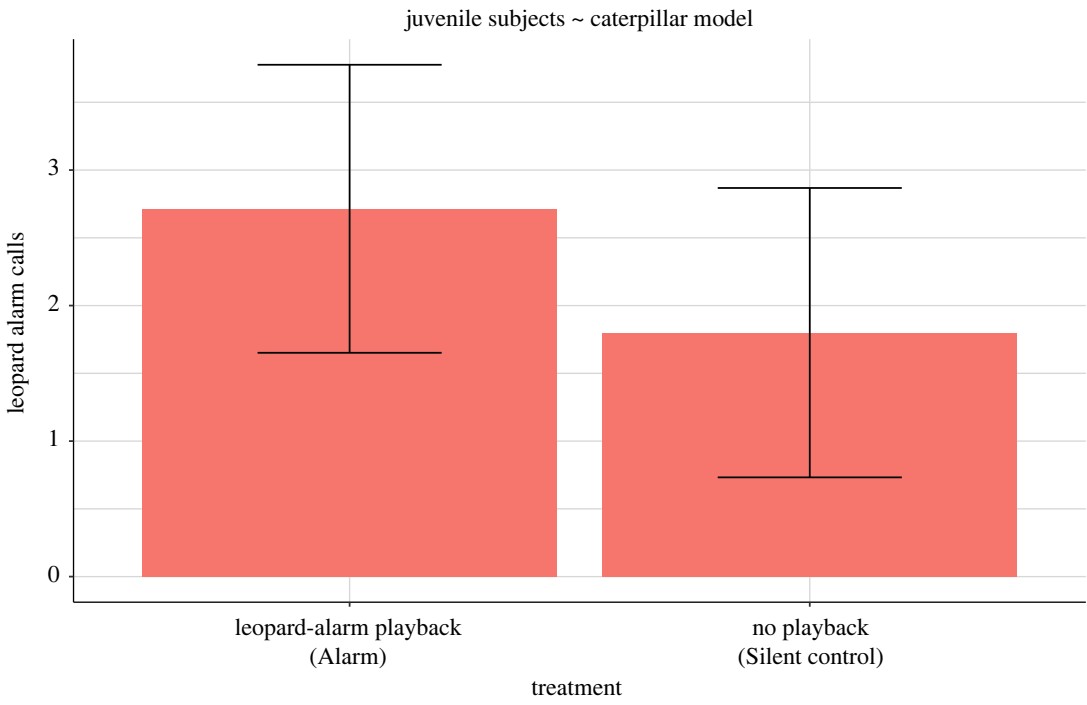

**Figure 6.** Mean (±s.e.) leopard alarms for the juvenile subjects exposed to the 'caterpillar' model.

adults ($n = 26$, model type: $\chi_1^2 = 4.92$, $p = 0.026$; age: $\chi_1^2 = 4.54$, $p = 0.032$; treatment: $\chi_1^2 = 0.01$, $p = 0.89$, $R^2 = 0.65$; figure 5), with most alarms given to the 'caterpillar' model by juveniles (68% of alarms; table 2).

In *post-hoc* analyses of individual assessments, we found that the 'Alarm' treatment elicited more leopard alarm calls than the 'Silent control' treatment (juveniles, 'caterpillar' model: $n = 12$, $\chi_1^2 = 8.69$, $p = 0.003$, $R^2 = 0.47$; figure 6). Leopard alarms also increased as the subjects' distance from the model increased ($n = 12$, $\chi_1^2 = 9.18$, $p = 0.002$). Snake calls, on the other hand, were not significantly affected, neither by treatment type nor by any other factor ($n = 12$, $\chi_1^2 = 0.11$, $p = 0.73$, $R^2 = 0.18$; electronic supplementary material, figure S4).

# 4. Discussion

## 4.1. Summary

Humans acquire language socially by observing and interacting with others [46]. We investigated whether social learning was equally important in how non-human primates acquire their communicative competence. We regarded this as a non-trivial problem owing to the fact that primate communication and human language differ drastically in scope, flexibility and power of the signal repertoire. While non-human primate communication is highly constrained to small repertoires of largely inflexible, species-specific signals, humans have evolved a second communication system resulting from the ability to precisely control and shape the vocal tracts and its acoustic output in highly intricate ways [1]. In the light of these differences, how do primates learn to associate their signals with the appropriate referents, in this case, a novel predator? When acquiring such competence, do primates employ social learning processes similar to how humans acquire language?

We found wild vervet monkeys that heard another's alarm calls while encountering an unfamiliar animal model exhibited a number of behavioural changes in how they reacted to the model in a future encounter. Compared to control subjects, monkeys showed increased predator inspection and corresponding alarm calling towards the model, although the latter effect was only seen in juveniles in only one type of alarm call. We tested two models and found that one (caterpillar) was more effective than the other (horse) in eliciting alarm calls. There was also a similar trend in predator inspection behaviour. This may be owing to several factors, including that the 'caterpillar' model was probably perceived as more animate and more similar to a familiar predator class (snakes) than the 'horse' model. Nevertheless, some juveniles produced leopard alarms to the 'caterpillar' model during individual assessments in the 'Alarm' condition.

The significant increase in predator inspection in the 'Alarm' condition, irrespective of age, suggests rapid comprehension learning. Subjects seem to have attributed novel meaning to existing alarms through social learning in a single exposure [25,47]. This capacity of non-human primates to form links between alarm calls and novel visual stimuli, even after a single exposure, is akin to fast mapping seen in children (also documented in a domestic dog [48,49]).

## 4.2. Adaptive significance of fast-mapping

Theoretical models predict that such rapid formations of call-context associations through social learning could be especially important in continually changing environments, where more standard models of learning requiring repeated exposures are less useful [17,50]. In the real world, changes in the predatory fauna can happen in fast and unpredictable ways [51]. For example, in the Mawana game reserve, where we conducted this study, common terrestrial predators of monkeys are caracals (*Caracal caracal*) and jackals (*Canis mesomelas*). However, a pack of African wild dogs (*Lycaon pictus*) and several subadult male lions (*Panthera leo leo*) have transiently visited Mawana over the years (A. Deshpande 2018, personal observation). This illustrates the adaptive value of inspection behaviour, granting individuals vital learning opportunities about predator behaviour and minimizing chances of making mistakes when responding to predators.

Our results on rapid comprehension learning are consistent with studies on fishes, birds, marsupials and other primate species [23,52–54]. Rapid learning is also essential during development. Infant primates are born into a local fauna that can vary drastically between locations, requiring individuals to categorize each animal species it encounters as either dangerous or harmless. Given the scale of the task, it is clear that acquiring this competence is highly unlikely without some form of social learning.

Recent computational models have suggested a similar mechanism of socially learned associative processes in the predator context [55]. Using existing nomenclature, stimulus enhancement may sufficiently account for the observed phenomenon [17,56]. Either way, individuals seem to acquire meaning and retain socially acquired predator knowledge from one single exposure.

## 4.3. Ontogenetic effects

Our *post hoc* analysis on the subset of juveniles exposed to the 'caterpillar' model shows that subjects elicited more leopard alarms if they have heard leopard alarms during the group exposure. This suggests usage learning of alarm calls by juveniles. However, not all juveniles equally produced

leopard alarms or even elicited an alarm call. This might reflect individual differences in learning [57]. On the other hand, it is also possible that call usage learning may be not as rapid as formations of call-context associations. For example, grey seals (*Halichoerus grypus*) learned to produce calls on command in captive settings; however, it took multiple trials [58]. This result suggests that social learning as a mechanism is not limited to the manipulative task but also helps to refine the usage of pre-existing alarm calls, at least in juveniles. Leopard alarm calls by juveniles for the 'caterpillar' model were also affected by the distance between the subject and the model at the start. We speculate that it reflects the perceived safety of the individuals before the alarm call. Alarm calling is a conspicuous behaviour and can attract predator attention; thus, a safe distance from the predator can reduce the associated risk [59]. Also, the juveniles alarm called less when they saw the model for a longer time. More model exposure might have provided the subjects with additional cues about the harmlessness of the model, which could have resulted in fewer alarm calls. Though, these speculations need to be examined empirically.

## 4.4. Core knowledge

It is also important to note that the 'caterpillar' model consistently elicited snake alarms across treatments, making it the 'default' alarm call for this type of model. Perhaps, monkeys perceived a resemblance with predators they were already familiar with (in this case, pythons) in terms of shape, movement pattern and encounter location, which might have biased categorization and alarm call type [60,61]. However, this suggestion is in contrast with the recent findings on congeneric green monkeys (*Chlorocebus sabaeus*). These primates were never observed alarm calling to birds of prey but instantly produced eagle-like alarm calls to an artificial flying drone, suggesting a hard-wired link between eagle alarms and aerial threat [27]. It is not possible to reject either of the two explanations from our results. Future experiments using ambiguous predator models of different predator categories in combination with playback of different alarm call types are needed to examine both possibilities.

The difference in alarm calls between the two predator models across the treatments and some indication of more inspection towards the 'caterpillar' model is noteworthy. The 'caterpillar' model consistently elicited heightened responses. Although both the models have eyes and are comparable in size, they differ in the aspects of animacy. The 'caterpillar' model made more enlivened movements with moving legs and was fluffier in texture, whereas the 'horse' model impassively dragged on the ground during exposure. It is possible that these differences in the animacy of the models could account for the differences in the anti-predatory responses. Similar results were found in captive blackbirds (*Turdus merula*), which were more scared of realistic-looking non-predatory bird models than the plastic bottles, and wild jackdaws (*Corvus monedula*), which were more attentive to the unexpected animacy of the objects [18,62]. In primates, including humans, the categorization of objects as animate or inanimate develops early in life or could even be innate. It constitutes what is known as 'initial' or 'core knowledge' [63,64]. Our results show that anti-predatory responses to the models seem to be based on integrating core knowledge and socially acquired knowledge [65].

In a subsidiary analysis, we compared differences between Alarm with Grunt control treatments, which revealed that predator inspection was specific following Alarm treatment. Unlike the grunts used in affiliative social interactions [34], leopard alarms might contain some information, such as the urgency or nature of the threat [66]. Alternatively, leopard alarms may not contain deterministic but probabilistic information about a threat that triggers visual search and, once located, leads to a formation of an association. The latter explanation of making pragmatic inference seems plausible since leopard alarm calls in this species are used in other multiple, potentially dangerous contexts, such as inter and intra-group aggression [31].

Although our results were as expected, the overall interpretation and conclusions remain preliminary owing to the incomplete 'Grunt control' condition. As explained, this was because we were forced to abandon the originally planned experimental design, owing to equipment malfunctioning. It resulted in an imbalance of treatments and reduction of sample size. Another issue is that, owing to the opportunistic selection of subjects, we were unable to control for age and sex class effects [67,68]. For future studies, we recommend taking appropriate precautions such as prioritizing treatments for the impact of invalid trials and hoping that results can eventually be replicated, including all control treatments.

## 5. Conclusion

Our results indicate that conspecific alarm calls facilitate learning about novel predators in wild primates. This finding corroborates Seyfarth & Cheney's pioneering observations that vervet monkeys acquire

aspects of anti-predatory behaviour by attending to the behaviour of others [26]. Our results also extend recent findings of rapid comprehension learning in congeneric green monkeys to the social learning realm [27]. Furthermore, our study expands the current theory by demonstrating the influence of other factors on social learning, notably age and core knowledge. Overall, competence in alarm call comprehension and production, it appears, is acquired before adulthood through social learning and by refining pre-existing cognitive biases. We have demonstrated that our experimental paradigm, although complex and vulnerable to invalid trials, is suitable for free-ranging animals and in all likelihood also useful for testing other species.

Ethics. Our study was approved by Ezemvelo KZN Wildlife and the Ethics Commission of the University of Neuchâtel (permit no. 28/2018).

Data accessibility. Data and the code for the analyses are available here: https://osf.io/ycnhj/?view_only= ade1f400289a46d18b6ef540bc7cbf8a. Data are also available in the electronic supplementary material [69].

Authors' contributions. A.D.: conceptualization, formal analysis, investigation, methodology, project administration, visualization, writing—original draft, writing—review and editing; B.v.B.: investigation, methodology, writing—review and editing; K.Z.: conceptualization, funding acquisition, project administration, supervision, writing—review and editing.

All authors gave final approval for publication and agreed to be held accountable for the work performed therein.

Conflict of interest declaration. The authors declare no competing interests.

Funding. This study primarily was funded by the Swiss National Science Foundation through grant nos. (31003A_166458 and 310030_185324) awarded to K.Z.

Acknowledgements. We thank Jean Robar for support with equipment, José van der Geest for acting as a second coder, Christof Neumann for suggestions on the experimental design, Radu Slobodeanu for support with statistical analyses, Erica van de Waal for granting access to the study groups, and all members of the Inkawu Vervet Project for support during fieldwork. The study was funded by the Swiss National Science Foundation.

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
