## [Peer Review File · Royal Society Open Science]

Review History

RSOS-210560.R0 (Original submission)

Review form: Reviewer 1

Is the manuscript scientifically sound in its present form?

No

Are the interpretations and conclusions justified by the results?

No

Is the language acceptable?

Yes

Do you have any ethical concerns with this paper?

No

Have you any concerns about statistical analyses in this paper?

Yes

Recommendation?

Reject

Comments to the Author(s)

This ms seeks to test if vervet monkeys have one-trial social learning about potential predators. The topic is timely, and particularly interesting in primates because of their advanced cognitive abilities yet limited vocal repertoire. Furthermore, the fieldwork was challenging and novel. The authors conclude that “call meaning can be acquired by one-trial social learning but that subject age and core knowledge about predators additionally moderate the acquisition of novel call-referent associations”.

I was not a reviewer on the original submission, to Proceedings B, but I share the concerns of the reviewers and editors of that submission. I believe that experimental limitations mean that it is difficult to draw conclusions from the results. My main concerns are as follows, and mostly echo the concerns of previous reviewers:

- 1) The main analysis compares the response to model animals after they had first seen the models associated either with alarm calls or silence. The experiment can therefore potentially draw conclusions about the effects of an associated sound, and not specifically about alarm calls. It is unclear whether any association is about potential “predators” or any objects. The experimental design did have another type of call (“grunt”), but those data were incomplete. In addition, grunts were recorded from subordinate males, whereas alarm calls were from adult males, so that the differences observed could potentially relate to the social standing of the caller, and not the type of call. (Also see point below about individual differences.)
- 2) The main data were not collected blind with respect to the hypothesis. There was a sample of videos scored blind, which showed a reasonable correlation with the main scoring, but it would be best to have all scored blind.
- 3) There could be more detail on methods. A) Playback amplitude was judged by ear to match the perceived amplitude of calls. Amplitude should be quantified, to allow replication by others. In this case, when comparing alarm calls with silence, the exact amplitude probably doesn’t matter, but it does potentially matter when comparing two types of calls, such as alarm calls and grunts, as was done in subsidiary analyses. B) It would be good to have more detail on the playback calls, and true replication. Were the calls of only one individual used per group for alarms and grunts? If so, responses to different individuals could confound apparent effects of call type.
- 4) I don’t understand the rationale of presenting two models. The work is about social learning, not about predator discrimination or different types of alarm call. This complexity seems to confuse matters, and leads to speculations about model types and call types that are not really warranted by the experimental design.

Despite these concerns, I think this work is interesting and could be used as a springboard for future work. The basic approach seems valuable, and many practical matters have been resolved. Impressive work, despite the limitations!

Review form: Reviewer 2

Is the manuscript scientifically sound in its present form?

No

Are the interpretations and conclusions justified by the results?

Yes

Is the language acceptable?

Yes

Do you have any ethical concerns with this paper?

No

Have you any concerns about statistical analyses in this paper?

No

Recommendation?

Major revision is needed (please make suggestions in comments)

Comments to the Author(s)

This paper is a resubmission of one initially submitted to Proc. Roy. Soc. B and the authors appear to have dealt with many of the issues raised by reviewers of the original manuscript. The Introduction and Methods are improved, and it has helped to remove the material on scratching. However, the main criticism of the initial reviewers remains- the fact that only one group was tested with another type of social sound. Both reviewers had emphasized the problem with using only silence as a control for alarm calls rather than a socially relevant sound unrelated to alarm. It is true that there was no effect on the small number of animals from a single group that were tested with grunts, suggesting that this might have worked for other groups, but we have only a single group which could be idiosyncratic. Although there is some mention of the missing grunt sessions in the Discussion the authors could still write a stronger caveat about their results. It is important that future researchers understand the problems with using silence as the only control stimulus and the importance of using appropriate auditory stimuli as controls in playback studies.

In the text Figure 6 is said to show effects of distance from the model, but the figure shown seems to show the leopard alarms given by juveniles to the caterpillar model instead. This figure is associated in the text with Figure 5. Since only two bars are shown, this figure could be reduced to a simple description in the text with means and variance provided. Figure 5 is cited appropriately in l. 310.

References still mix capitalizing all major words in titles with capitalizing only the first word. References 36, 38, 45 and 50 have incomplete citations so the authors have not dealt with the references as they claim to have.

l. 198 "Adult females"

Review form: Reviewer 3

Is the manuscript scientifically sound in its present form?

Yes

Are the interpretations and conclusions justified by the results?

Yes

Is the language acceptable?

Yes

Do you have any ethical concerns with this paper?

No

Have you any concerns about statistical analyses in this paper?

No

Recommendation?

Accept with minor revision (please list in comments)

Comments to the Author(s)

I have reviewed the paper "One trial social learning of vervet monkey alarm calling" by Deshpande et al.

This manuscript aims to unpack the cognitive mechanisms underlying the comprehension and usage of alarm calls in monkeys with an ultimate goal of bridging the gap in understanding between human and primate vocal flexibility.

Through conducting presentation and playback experiments in the wild, the authors demonstrate that coupling unfamiliar predator models with alarm vocalisations leads to heightened predator inspection compared to controls where alarm calls were not presented. Since this was a single shot learning scenario, the authors argue meaning of alarm vocalisations in vervets can be acquired via social learning with the absolute minimum of exposure (one trial).

In general I thought this was an very interesting paper that engages in a suite of experiments that are extremely difficult to conduct under natural conditions and hence the data is both insightful and valuable to the field. Previous reviews (at Proc B) flagged the importance of a non-alarm control and critiqued the small sample size associated with this, however I think these critiques overlook the basic findings that there is clearly an influence of conspecific vocal presence on alarm call comprehension learning and fail to appreciate the fact that the authors are working with a wild primate population where field experiments come with a whole host of complications and constraints. These would be easier to circumvent in captivity, but the ecological and evolutionary relevance which is so central to the authors' questions would be lacking and in my opinion this would be far more problematic.

This in conjunction with the fact that there is previous evidence suggesting primates (and indeed other species) are capable of single exposure learning of call/word meaning (albeit using different paradigms than the one presented here) increases the robustness of the author's findings.

I think this paper fits very well into the remit of RSOS and I would recommend publication following the addressing of a few issues here and there:

1. L46: I think the term "conscious" is maybe a little strong here. Even though I appreciate the authors are not suggesting chimps are conscious beings, I'm sure many people in the field would predictably take issue with this term and hence it might be safer to leave this term away!
2. L191: I couldn't immediately find whether the authors kept track of the location of the call provider during playbacks? If so, did they notice any responses of that individual to the playbacks themselves?
3. L224-233: I applaud the authors here for full transparency regarding the logistical challenges that come with conducting such experiments in the field and ultimately the downstream consequences of that on experimental design etc.
4. L275: Maybe clarify here you are referring to "presentation model" since this could be confused with "statistical model".
5. L315: Can the authors more precisely clarify the directionality of this effect (i.e. were more leopard alarm calls produced when the subject was closer or further away?).
6. L366: remove "to obtain".
7. L410: Change model to "models".

Decision letter (RSOS-210560.R0)

Dear Dr Deshpande

The Editors assigned to your paper RSOS-210560 "One-trial social learning of vervet monkey alarm calling" have now received comments from reviewers and would like you to revise the paper in accordance with the reviewer comments and comments from the Associate Editor. Please note this decision does not guarantee eventual acceptance.

Please submit your revised manuscript and required files (see below) no later than 21 days from today's (ie 27-Jun-2022) date. Note: the ScholarOne system will 'lock' if submission of the revision is attempted 21 or more days after the deadline. If you do not think you will be able to meet this deadline please contact the editorial office immediately.

on behalf of Dr Jennifer Cook (Associate Editor) and Essi Viding (Subject Editor)
openscience@royalsociety.org

Associate Editor Comments to Author (Dr Jennifer Cook):

Associate Editor: 1

Comments to the Author:

Dear Dr Deshpande,

Your manuscript has now been seen by three reviewers. All reviewers were positive about aspects of the work and we would like to invite you to revise and resubmit your manuscript. However, please note that there were a number of points of concern. Most notably, all reviewers were concerned about the silent control stimulus. We suggest that you exercise more caution regarding interpretation of your results and add a fuller explanation of the problems with using

silence as the only control stimulus such that future researchers can appreciate the importance of using appropriate auditory controls in playback studies.

Very best wishes,
Jennifer

Reviewer comments to Author:

Reviewer: 1

Comments to the Author(s)

This ms seeks to test if vervet monkeys have one-trial social learning about potential predators. The topic is timely, and particularly interesting in primates because of their advanced cognitive abilities yet limited vocal repertoire. Furthermore, the fieldwork was challenging and novel. The authors conclude that “call meaning can be acquired by one-trial social learning but that subject age and core knowledge about predators additionally moderate the acquisition of novel call-referent associations”.

I was not a reviewer on the original submission, to Proceedings B, but I share the concerns of the reviewers and editors of that submission. I believe that experimental limitations mean that it is difficult to draw conclusions from the results. My main concerns are as follows, and mostly echo the concerns of previous reviewers:

- 1) The main analysis compares the response to model animals after they had first seen the models associated either with alarm calls or silence. The experiment can therefore potentially draw conclusions about the effects of an associated sound, and not specifically about alarm calls. It is unclear whether any association is about potential “predators” or any objects. The experimental design did have another type of call (“grunt”), but those data were incomplete. In addition, grunts were recorded from subordinate males, whereas alarm calls were from adult males, so that the differences observed could potentially relate to the social standing of the caller, and not the type of call. (Also see point below about individual differences.)
- 2) The main data were not collected blind with respect to the hypothesis. There was a sample of videos scored blind, which showed a reasonable correlation with the main scoring, but it would be best to have all scored blind.
- 3) There could be more detail on methods. A) Playback amplitude was judged by ear to match the perceived amplitude of calls. Amplitude should be quantified, to allow replication by others. In this case, when comparing alarm calls with silence, the exact amplitude probably doesn’t matter, but it does potentially matter when comparing two types of calls, such as alarm calls and grunts, as was done in subsidiary analyses. B) It would be good to have more detail on the playback calls, and true replication. Were the calls of only one individual used per group for alarms and grunts? If so, responses to different individuals could confound apparent effects of call type.
- 4) I don’t understand the rationale of presenting two models. The work is about social learning, not about predator discrimination or different types of alarm call. This complexity seems to confuse matters, and leads to speculations about model types and call types that are not really warranted by the experimental design.

Despite these concerns, I think this work is interesting and could be used as a springboard for future work. The basic approach seems valuable, and many practical matters have been resolved. Impressive work, despite the limitations!

Reviewer: 2

Comments to the Author(s)

This paper is a resubmission of one initially submitted to Proc. Roy. Soc. B and the authors appear to have dealt with many of the issues raised by reviewers of the original manuscript. The Introduction and Methods are improved, and it has helped to remove the material on scratching.

However, the main criticism of the initial reviewers remains- the fact that only one group was tested with another type of social sound. Both reviewers had emphasized the problem with using only silence as a control for alarm calls rather than a socially relevant sound unrelated to alarm. It is true that there was no effect on the small number of animals from a single group that were tested with grunts, suggesting that this might have worked for other groups, but we have only a single group which could be idiosyncratic. Although there is some mention of the missing grunt sessions in the Discussion the authors could still write a stronger caveat about their results. It is important that future researchers understand the problems with using silence as the only control stimulus and the importance of using appropriate auditory stimuli as controls in playback studies.

In the text Figure 6 is said to show effects of distance from the model, but the figure shown seems to show the leopard alarms given by juveniles to the caterpillar model instead. This figure is associated in the text with Figure 5. Since only two bars are shown, this figure could be reduced to a simple description in the text with means and variance provided. Figure 5 is cited appropriately in l. 310.

References still mix capitalizing all major words in titles with capitalizing only the first word. References 36, 38, 45 and 50 have incomplete citations so the authors have not dealt with the references as they claim to have.

l. 198 "Adult females"

Reviewer: 3

Comments to the Author(s)

I have reviewed the paper "One trial social learning of vervet monkey alarm calling" by Deshpande et al.

This manuscript aims to unpack the cognitive mechanisms underlying the comprehension and usage of alarm calls in monkeys with an ultimate goal of bridging the gap in understanding between human and primate vocal flexibility.

Through conducting presentation and playback experiments in the wild, the authors demonstrate that coupling unfamiliar predator models with alarm vocalisations leads to heightened predator inspection compared to controls where alarm calls were not presented. Since this was a single shot learning scenario, the authors argue meaning of alarm vocalisations in vervets can be acquired via social learning with the absolute minimum of exposure (one trial).

In general I thought this was an very interesting paper that engages in a suite of experiments that are extremely difficult to conduct under natural conditions and hence the data is both insightful and valuable to the field. Previous reviews (at Proc B) flagged the importance of a non-alarm control and critiqued the small sample size associated with this, however I think these critiques overlook the basic findings that there is clearly an influence of conspecific vocal presence on alarm call comprehension learning and fail to appreciate the fact that the authors are working with a wild primate population where field experiments come with a whole host of complications and constraints. These would be easier to circumvent in captivity, but the ecological and evolutionary relevance which is so central to the authors' questions would be lacking and in my opinion this would be far more problematic.

This in conjunction with the fact that there is previous evidence suggesting primates (and indeed other species) are capable of single exposure learning of call/word meaning (albeit using different paradigms than the one presented here) increases the robustness of the author's findings.

I think this paper fits very well into the remit of RSOS and I would recommend publication following the addressing of a few issues here and there:

1. L46: I think the term “conscious” is maybe a little strong here. Even though I appreciate the authors are not suggesting chimps are conscious beings, I’m sure many people in the field would predictably take issue with this term and hence it might be safer to leave this term away!
2. L191: I couldn’t immediately find whether the authors kept track of the location of the call provider during playbacks? If so, did they notice any responses of that individual to the playbacks themselves?
3. L224-233: I applaud the authors here for full transparency regarding the logistical challenges that come with conducting such experiments in the field and ultimately the downstream consequences of that on experimental design etc.
4. L275: Maybe clarify here you are referring to “presentation model” since this could be confused with “statistical model”.
5. L315: Can the authors more precisely clarify the directionality of this effect (i.e. were more leopard alarm calls produced when the subject was closer or further away?).
6. L366: remove “to obtain”.
7. L410: Change model to “models”.

===PREPARING YOUR MANUSCRIPT===

If you have been asked to revise the written English in your submission as a condition of publication, you must do so, and you are expected to provide evidence that you have received language editing support. The journal would prefer that you use a professional language editing service and provide a certificate of editing, but a signed letter from a colleague who is a fluent speaker of English is acceptable. Note the journal has arranged a number of discounts for authors using professional language editing services (<https://royalsociety.org/journals/authors/benefits/language-editing/>).

===PREPARING YOUR REVISION IN SCHOLARONE===

Author's Response to Decision Letter for (RSOS-210560.R0)

See Appendix A.

Decision letter (RSOS-210560.R1)

Dear Dr Deshpande:

I am pleased to inform you that your manuscript entitled "Preliminary evidence for one-trial social learning of vervet monkey alarm calling" is now accepted for publication in Royal Society Open Science.

Please remember to make any data sets or code libraries 'live' prior to publication, and update any links as needed when you receive a proof to check - for instance, from a private 'for review' URL to a publicly accessible 'for publication' URL. It is also good practice to add data sets, code and other digital materials to your reference list.

Royal Society Open Science is a fully open access journal. A payment may be due before your article is published. Our partner Copyright Clearance Center's RightsLink for Scientific Communications will contact the corresponding author about your open access options from the email domain @copyright.com (if you have any queries regarding fees, please see <https://royalsocietypublishing.org/rsos/charges> or contact authorfees@royalsociety.org).

on behalf of Dr Jennifer Cook (Associate Editor) and Dr Essi Viding (Subject Editor).

<https://www.facebook.com/RoyalSocietyPublishing/>

Appendix A

Dear Dr Cook,

We are grateful to you and the reviewers for the detailed responses on our article (RSOS-210560). We have considered all suggestions and incorporated all of them in the revised manuscript, submitted along with this letter.

Best wishes,

Adwait Deshpande (on behalf of all authors)

Response to the main issue raised by the editor and all the reviewers:

The common issue raised by the editor and all the reviewers are related to the lack of a 'grunt' control condition, a limitation of this study. As suggested, we have therefore made our interpretations less assertive and pointed out the preliminary nature of our conclusions in the title of the manuscript, which is further elaborated in the discussion. We also provide suggestions for future research that could build on the current design and data in ecologically relevant settings.

Below are the detailed responses to the editor's and reviewers' comments. Line numbers refer to the revised version (track changes mode) of the manuscript.

Associate Editor: 1

Dear Dr Deshpande,

Your manuscript has now been seen by three reviewers. All reviewers were positive about aspects of the work and we would like to invite you to revise and resubmit your manuscript. However, please note that there were a number of points of concern. Most notably, all reviewers were concerned about the silent control stimulus. We suggest that you exercise more caution regarding interpretation of your results and add a fuller explanation of the problems with using silence as the only control stimulus such that future researchers can appreciate the importance of using appropriate auditory controls in playback studies.

*Very best wishes,
Jennifer*

We thank the associate editor for her positive overall assessment. We agree that the silent control condition should have been complemented with a further grunt control condition. We address this point explicitly in the revised version, combined with recommendations for future research.

Reviewer: 1

Comments to the Author(s): This ms seeks to test if vervet monkeys have one-trial social learning about potential predators. The topic is timely, and particularly interesting in primates because of their advanced cognitive abilities yet limited vocal repertoire. Furthermore, the fieldwork was challenging and novel. The authors conclude that "call meaning can be acquired by one-trial social learning but that subject age and core knowledge about predators additionally moderate the acquisition of novel call-referent associations".

I was not a reviewer on the original submission, to Proceedings B, but I share the concerns of the reviewers and editors of that submission. I believe that experimental limitations mean that it is difficult to draw conclusions from the results. My main concerns are as follows, and mostly echo the concerns of previous reviewers:

1) The main analysis compares the response to model animals after they had first seen the models associated either with alarm calls or silence. The experiment can therefore potentially draw conclusions about the effects of an associated sound, and not specifically about alarm calls. It is unclear whether any association is about potential "predators" or any objects. The experimental design did have another type of call ("grunt"), but those data were incomplete.

We are aware of the fact that the lack of a 'grunt' control is a significant limitation (see our response above to all reviewers), which we fully acknowledge in the revised version.

In addition, grunts were recorded from subordinate males, whereas alarm calls were from adult males, so that the differences observed could potentially relate to the social standing of the caller, and not the type of call. (Also see point below about individual differences.)

The recording was from a subordinate adult male, now explained in the manuscript (**L 140-147**). We agree that, in future experiments, the social status of the call provider should be taken into account, as this may affect social learning by others.

2) The main data were not collected blind with respect to the hypothesis. There was a sample of videos scored blind, which showed a reasonable correlation with the main scoring, but it would be best to have all scored blind.

We agree that having all videos scored blindly would have been ideal. In our study, this would have been unreasonably onerous because for, for each trial, it was necessary to explain to the rater the exact spatial position of the hide and model, relative to the subject's head position, in order to make a meaningful judgements of whether predator inspection took place. We therefore decided to follow the standard procedure of a second rater scoring 20% of the material, which led to an acceptable interrater agreement.

3) There could be more detail on methods. A) Playback amplitude was judged by ear to match the perceived amplitude of calls. Amplitude should be quantified, to allow replication by others. In this case, when comparing alarm calls with silence, the exact amplitude probably doesn't matter, but it does potentially matter when comparing two types of calls, such as alarm calls and grunts, as was done in subsidiary analyses.

We agree that providing amplitude data under field conditions would have added further methodological rigour. However, we decided against this because each trial differed in terms of density of local vegetation and distance between speaker and subject. In addition, multiple playback studies have shown that, as long as monkeys can hear a playback call clearly, differences in amplitude never had an impact on their responses. In our study, the amplitude of leopard alarm calls was much higher compared to the soft 'grunts', a close-range social call, suggesting that call amplitude is very unlikely to have an impact.

B) It would be good to have more detail on the playback calls, and true replication. Were the calls of only one individual used per group for alarms and grunts? If so, responses to different individuals could confound apparent effects of call type.

We have clarified how many calls of different individuals were used in each group and experimental blocks (**L 140-147**).

4) I don't understand the rationale of presenting two models. The work is about social learning, not about predator discrimination or different types of alarm call. This complexity seems to confuse matters and leads to speculations about model types and call types that are not really warranted by the experimental design.

We agree that, in principle, one model type would have been sufficient to address our question. However, as we were using unfamiliar terrestrial models with no clear resemblance to any of their natural predators, we simply wanted to increase the certitude that the physical characteristic of the model had no impact on the monkeys' responses. Importantly, our results showed that differences in animacy of the two models had an effect, which we regard as important information for future similar experiments.

Despite these concerns, I think this work is interesting and could be used as a springboard for future work. The basic approach seems valuable, and many practical matters have been resolved. Impressive work, despite the limitations!

We are grateful for the reviewer's overall positive assessment, despite the shortcomings in the design.

Reviewer: 2

Comments to the Author(s)

This paper is a resubmission of one initially submitted to Proc. Roy. Soc. B and the authors appear to have dealt with many of the issues raised by reviewers of the original manuscript. The Introduction and Methods are improved, and it has helped to remove the material on scratching. However, the main criticism of the initial reviewers remains- the fact that only one group was tested with another type of social sound. Both reviewers had emphasized the problem with using only silence as a control for alarm calls rather than a socially relevant sound unrelated to alarm. It is true that there was no effect on the small number of animals from a single group that were tested with grunts, suggesting that this might have worked for other groups, but we have only a single group which could be idiosyncratic. Although there is some mention of the missing grunt sessions in the Discussion the authors could still write a stronger caveat about their results. It is important that future researchers understand the problems with using silence as the only control stimulus and the importance of using appropriate auditory stimuli as controls in playback studies.

We thank the reviewer for his/her assessment, which very much echoes the responses of the other reviewers. We have addressed and highlighted the limitations throughout the manuscript, highlighting that conclusions remain preliminary.

In the text Figure 6 is said to show effects of distance from the model, but the figure shown seems to show the leopard alarms given by juveniles to the caterpillar model instead. This figure is associated in the text with Figure 5. Since only two bars are shown, this figure could be reduced to a simple description in the text with means and variance provided. Figure 5 is cited appropriately in l. 310.

Thank you for pointing out this error, which we have now corrected (**L 298-300**). We have decided to retain Figure 6 in the manuscript as we discuss these results briefly in the Discussion.

References still mix capitalizing all major words in titles with capitalizing only the first word. References 36, 38, 45 and 50 have incomplete citations so the authors have not dealt with the references as they claim to have.

We have corrected these problems.

l. 198 "Adult females"

Corrected (**L 187**)

Reviewer: 3

Comments to the Author(s)

I have reviewed the paper "One trial social learning of vervet monkey alarm calling" by Deshpande et al. This manuscript aims to unpack the cognitive mechanisms underlying the comprehension and usage of alarm calls in monkeys with an ultimate goal of bridging the gap in understanding between human and primate vocal flexibility. Through conducting presentation and playback experiments in the wild, the authors demonstrate that coupling unfamiliar predator models with alarm vocalisations leads to heightened predator inspection compared to controls where alarm calls were not presented. Since this was a single shot learning scenario, the authors argue meaning of alarm vocalisations in vervets can be acquired via social learning with the absolute minimum of exposure (one trial).

In general I thought this was an very interesting paper that engages in a suite of experiments that are extremely difficult to conduct under natural conditions and hence the data is both insightful and valuable to the field. Previous reviews (at Proc B) flagged the importance of a non-alarm control and critiqued the small sample size associated with this, however I think these critiques overlook the basic findings that there is clearly an influence of conspecific vocal presence on alarm call comprehension learning and fail to appreciate the fact that the authors are working with a wild primate population where field experiments come with a whole host of complications and constraints. These would be easier to circumvent in captivity, but the ecological and evolutionary relevance which is so central to the authors' questions would be lacking and in my opinion this would be far more problematic.

This in conjunction with the fact that there is previous evidence suggesting primates (and indeed other species) are capable of single exposure learning of call/word meaning (albeit using different paradigms than the one presented here) increases the robustness of the author's findings.

I think this paper fits very well into the remit of RSOS and I would recommend publication following the addressing of a few issues here and there:

1. L46: I think the term "conscious" is maybe a little strong here. Even though I appreciate the authors are not suggesting chimps are conscious beings, I'm sure many people in the field would predictably take issue with this term and hence it might be safer to leave this term away!

We agree with the reviewer's view and have removed the term (**L 45**)

2. L191: I couldn't immediately find whether the authors kept track of the location of the call provider during playbacks? If so, did they notice any responses of that individual to the playbacks themselves?

During the group exposure trial, we made sure the call provider was not present in the immediate vicinity of the group. Given the complex setup, it was logistically impossible to allocate another observer to track the call provider's reaction. However, given that the data set consisted of single trials per group, the call provider's (eventual) reaction to its own calls had no impact on the dataset.

3. L224-233: *I applaud the authors here for full transparency regarding the logistical challenges that come with conducting such experiments in the field and ultimately the downstream consequences of that on experimental design etc.*

Thank you very much for this comment. We have highlighted the limitations of this study and provided recommendations for other researchers for similar future experiments. **(L 403-415)**

4. L275: *Maybe clarify here you are referring to "presentation model" since this could be confused with "statistical model".*

Rectified **(L 259)**

5. L315: *Can the authors more precisely clarify the directionality of this effect (i.e. were more leopard alarm calls produced when the subject was closer or further away?).*

We have now rephrased the sentence **(L 297-299)**

6. L366: *remove "to obtain".*

Removed **(L 345)**

7. L410: *Change model to "models".*

This sections is now rephrased